# PointPatchRL - Masked Reconstruction Improves Reinforcement Learning on Point Clouds

**Balázs Gyenes[1,2], Nikolai Franke[1], Philipp Becker[1], Gerhard Neumann[1]**
[1]Institute for Anthropomatics and Robotics, Karlsruhe Institute of Technology, Germany
[2]HIDSS4Health - Helmholtz Information and Data Science School for Health,
Karlsruhe/Heidelberg, Germany

**Abstract:** Perceiving the environment via cameras is crucial for Reinforcement Learning (RL) in robotics. While images are a convenient form of representation, they often complicate extracting important geometric details, especially with varying geometries or deformable objects. In contrast, point clouds naturally represent this geometry and easily integrate color and positional data from multiple camera views. However, while deep learning on point clouds has seen many recent successes, RL on point clouds is under-researched, with only the simplest encoder architecture considered in the literature. We introduce PointPatchRL (PPRL), a method for RL on point clouds that builds on the common paradigm of dividing point clouds into overlapping patches, tokenizing them, and processing the tokens with transformers. PPRL provides significant improvements compared with other point-cloud processing architectures previously used for RL. We then complement PPRL with masked reconstruction for representation learning and show that our method outperforms strong model-free and model-based baselines on image observations in complex manipulation tasks containing deformable objects and variations in target object geometry. Videos and code are available at alrhub.github.io/pprl-website.

**Keywords:** Point Clouds, Self-Supervised Learning, Reinforcement Learning

## 1 Introduction

In recent years, Reinforcement Learning (RL) for robots enabled solving increasingly complex tasks [1, 2, 3]. A key factor contributing to these advancements is the development of representation learning techniques for encoding high-dimensional sensor data such as camera images or depth maps into compact, low-dimensional representations [4, 5, 6]. There are a variety of learning objectives to train visual representations for RL, such as image reconstruction [7], multi-view contrastive loss [8], or multi-view consistency [9]. However, these methods struggle to extract 3D geometric information about the scene and leave the handling of depth measurements under-explored. Not being able to provide the policy with a 3D inductive bias for resolving ambiguities due to occlusions or camera movement presents challenges for representations learned purely from pixels [10]. Point clouds provide a natural representation of 3D geometry and can straightforwardly combine color and positional information. In contrast to depth images, point clouds leverage the camera parameters to obtain a common geometric representation from one or more camera views, alleviating the issues of pixel-based approaches. While this requires measuring the relative poses between cameras, in many cases this can be done without tracking markers or SLAM algorithms. For example, if all cameras are mounted on the moving robot, the forward kinematics alone are sufficient. In addition, policies trained in simulation on point clouds instead of color images may be easier transferable to the real world [2, 11], since shapes are inherently easier to simulate than textures.

8th Conference on Robot Learning (CoRL 2024), Munich, Germany.

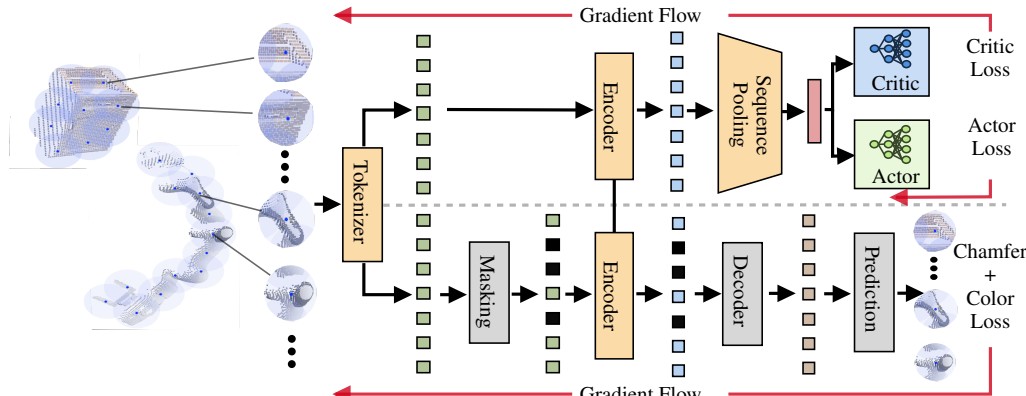

Figure 1: Schematic of PointPatchRL(PPRL) with an auxiliary masked reconstruction loss trained end-to-end for RL. **Top**: We train a patching-based tokenizer and transformer encoder to compute a latent embedding for the RL policy and state-action-value estimation using sequence pooling. The entire pipeline learns using the critic's gradients while we detach the latent embedding before providing it to the actor. **Bottom:** We augment the policy learning using masked reconstruction. Using the token sorting, masking, and transformer decoder introduced by PointGPT [17], we minimize the Chamfer distance for the point's positions and the mean squared reconstruction error for colors. This auxiliary loss provides an additional training signal for the shared encoder and tokenizer, improving RL performance and sample efficiency.

Since the introduction of PointNet [12] for processing raw point clouds directly, the number of methods in this field has grown considerably [13, 14, 15, 16, 17]. However, the use of these encoder architectures in the context of reinforcement learning is relatively under-explored.

In this work, we introduce PointPatchRL (PPRL), which builds on recent patching-based methods for point clouds such as Point-BERT [15], Point-MAE [16] and PointGPT [17]. These methods first divide the point clouds into overlapping patches, which are then tokenized and processed by a Transformer encoder [18]. By following this approach, PPRL can leverage the intrinsic geometric properties of point clouds to learn dense representations that capture the task-relevant features of the environment directly from raw point cloud data. We show that providing this representation to a Soft Actor-Critic [19] agent and training the encoder and tokenizer jointly with the critic already gives successful results compared to most point cloud-based and image-based baselines. We then extend PPRL with components from PointGPT [17] and introduce a masked reconstruction objective to increase the geometric information encoded in the latent representation. This modification requires adapting the padding and matching procedure to support point clouds of any size and integrating point-level color features. We demonstrate that PPRL combined with this auxiliary loss further improves performance, especially in more challenging mobile manipulation environments and tasks with varying object geometries and moving cameras. A schematic overview of our architecture and learning pipeline is given in Figure 1.

To summarize our main contributions: **(i)** we introduce PPRL, a novel, patching-based approach for RL on point clouds. **(ii)** We augment PPRL using a masked reconstruction learning objective that further significantly improves the performance of our agents. **(iii)** We empirically demonstrate the capability of our method to solve complex simulated robotics tasks conditioned on color and depth information relative to other point cloud-based and image-based approaches.

## 2 Related Work

**Image Representations for RL.** Images are a natural and versatile observation modality when using Reinforcement Learning (RL) in unstructured environments. However, naively using images as inputs to actor and critic networks often results in poor sample complexity, poor performance, or

both. Previous works propose sophisticated approaches such as image augmentation [20, 21, 22], autoencoding-based reconstruction [23, 7, 4], or other self-supervised methods [24, 25]. Driess et al. [9] learn a representation with the help of a multi-view consistency objective using Neural Radiance Fields (NeRF). Of particular importance to our work are approaches using masked autoencoding [26] on images for RL. Xiao et al. [5] use a pre-trained masked autoencoder for on-policy RL. Seo et al. [27] train a Dreamer world model [7] in the latent space of a masked autoencoder. They show that naively masking image patches can be suboptimal for RL and instead mask convolutional features. In subsequent work, they extend their approach to handle multiple viewpoints and make it robust to viewpoint perturbations [28], properties which are shared implicitly by our point cloud-based approach. While these approaches can be extended to depth images, we argue that point clouds are a more natural representation of the scene's geometry, as they leverage the camera geometry and straightforwardly merge points from multiple cameras into the same coordinate frame.

**Point Cloud Representation Learning.** Since the advent of PointNet [12], many neural network architectures for point clouds have been developed, mainly for classification and segmentation tasks. PointNet++ [13] improves upon PointNet by using the Farthest Point Sampling algorithm to hierarchically partition the point cloud. PointTransformer [14] introduces a transformer backbone for point cloud processing, iteratively applying vector attention on point-level features within a small neighborhood and downsampling similar to PointNet++. Point-MAE [16] adapts the popular masked autoencoding paradigm to learn point cloud embeddings, dividing the point cloud into patches, masking them, and using a transformer to reconstruct the masked patches. The network is trained to minimize the Chamfer distance between the reconstruction and the ground truth in the masked point patches. Point-BERT [15] uses block masking instead of random masking and predicts discrete point tokens instead of continuous ones. Point-GPT [17] addresses several weaknesses in Point-MAE and leverages the successful auto-regressive inference paradigm used in natural language processing. Despite their success, many of these architectures have never been used for RL, and we show that using the more modern point patching approach with an auxiliary objective as in PointGPT [17] improves performance on challenging (mobile) manipulation tasks.

**RL with Point Clouds.** Despite recent advances in point cloud processing, RL on point clouds for robotics is under-researched, with the most common application being dexterous manipulation of objects [29, 3, 30]. Huang et al. [29] use a pretrained PointNet feature extractor to learn a general dexterous manipulation policy capable of in-hand orientation of various objects. Chen et al. [3] train a sparse 3D CNN feature extractor end-to-end using a student-teacher learning method, where the teacher has access to privileged state information, while the student only relies on point clouds and proprioception. Wu et al. [30] use imitation learning to solve the dexterous manipulation task, also employing a pretrained PointNet that is finetuned during behavior cloning. Liu et al. [31] explore the impact of different coordinate frames on tasks from `ManiSkill2` [32] with point cloud observations combined with proprioception. Ling et al. [33] study the effectiveness of point cloud-based RL compared to image-based RL on a mix of two-dimensional and three-dimensional environments. They find that point cloud observations offer benefits over image observations in environments where understanding agent-object and object-object relationships is important and that a PointNet encoder outperforms a 3D SparseConvNet [34]. Wan et al. [35] make use of PointNet+Transformer [36], but this requires a segmented point cloud, which is not always available. Peri et al. [37] investigate the use of an encoder based on PointConv [38] for RL in latent imagination. However, a systematic comparison of point cloud encoder architectures is missing in the literature. In particular, there are no works that make use of the point patching paradigm, which we show empirically to be a powerful approach on complex 3D (mobile) manipulation tasks.

## 3 PointPatchRL

PointPatchRL (PPRL) assumes point clouds $\boldsymbol{X} \in \mathbb{R}^{m \times 3}$ with $m$ points. Here, the size $m$ can vary between the individual point clouds $\boldsymbol{X}$ encountered during RL and we thus design an encoder to compute a fixed-length embedding containing all task-relevant information provided by $\boldsymbol{X}$. We then provide this embedding to an actor and a critic which we train using a soft actor-critic (SAC) [19].

Our point cloud encoder is based on the point patching paradigm [15, 16, 17] with a transformer backbone, adapted to handle point clouds of arbitrary size. If additional color information is available for the points, we can include it by appending corresponding features to each point. We use a sequence pooling layer to obtain the fixed-length embedding required for RL. Besides training the encoder solely using the critic loss, we consider augmenting training by explicit representation learning. We do so using PointGPT [17], which learns representations via the reconstruction of masked point patches. During training, we separately compute the reconstruction loss with masking and the RL loss based on an embedding without masking, as illustrated in Figure 1.

## 3.1 Point Patch RL (PPRL)

**Tokenization.** To process the point cloud $X$ using a transformer, we first need to tokenize it. Here, grouping many points into a single token is desirable, as point clouds have low information density. Thus, we sample $n$ points as centroids $C = \text{fps}(X)$, $\quad C \in \mathbb{R}^{n \times 3}$ using Farthest Point Sampling (FPS) [13], which ensures good coverage and an even distribution of the centroids. Subsequently, we conduct a K-Nearest Neighbours (k-NN) search to select the $k$ nearest points of each centroid to obtain (potentially overlapping) point patches $P = \text{knn}(X, C)$, $\quad P \in \mathbb{R}^{n \times k \times 3}$. The coordinates of all points in a patch are normalized relative to the centroid's position. A lightweight version of PointNet [12], consisting of two MLPs and two max-pooling layers, converts point matches $P$ into input tokens $T \in \mathbb{R}^{n \times D}$ of size $D$.

**Transformer Architecture.** A standard transformer encoder receives the sequence of tokens $T$ and produces a new sequence of encoded tokens. To this end, we also need to encode the position of each centroid and add it to the respective token. As in PointGPT, we add sinusoidal positional encodings of the centroids' locations to the tokens.

**Reinforcement Learning.** We utilize soft-actor critic (SAC) which is a popular algorithm due to its sample efficiency. SAC learns a policy and a state-action value function network, which both require a fixed-length embedding as input vectors. We use a sequence pooling layer [39] to obtain this embedding. This layer computes a weighted sum over all input tokens, where the weight of each token is a function of that token, i.e.,

$$T^{\text{pooled}} = w^T T \quad \text{with} \quad w = \text{softmax}(g(T)) \quad \text{and} \quad T^{\text{pooled}} \in \mathbb{R}^d.$$

where $g$ is a linear layer. Compared with using an explicit query token to compute a fixed-length embedding, sequence pooling can improve the performance of small transformer networks [39].

If low-dimensional state observations are also available, they are projected with a linear layer to the same dimensionality as the point cloud embedding, and then concatenated with the embedding. Actor and critic networks share a single point cloud encoder, which is updated only with the gradients from the critic loss, as is standard practice in off-policy RL [24, 22].

## 3.2 Representation Learning Using Masked Reconstruction

We extend PPRL with a masked reconstruction objective, to benefit from an additional geometric learning signal. By adding sorting, masking, and a transformer decoder to our architecture, we obtain a modified version of PointGPT. Unless noted, all details below follow PointGPT.

After tokenization, the tokens are sorted according to the Morton order [41] (Z-order curve), which maps n-dimensional data to one dimension while preserving the locality of the data points. Because nearby point patches remain close to each other after sorting, the model can rely on neighboring tokens to inform the reconstruction. In contrast to the transformer encoder, the transformer decoder does not receive the encodings of the absolute positions of point patches, but only the relative direction between subsequent patches is encoded, i.e.,

$$E_i^r = \phi \left( \frac{C_i^s - C_{i-1}^s}{\| C_i^s - C_{i-1}^s \|_2} \right), \text{ for } i \in \{2, \ldots, n\}, \text{ and } E_1^r = C_1^S$$

where $C_i^s$ is the $i$th centroid of the ordered list of centroids. This avoids leaking global information about the point cloud structure through the positional embeddings to the decoder. The masking

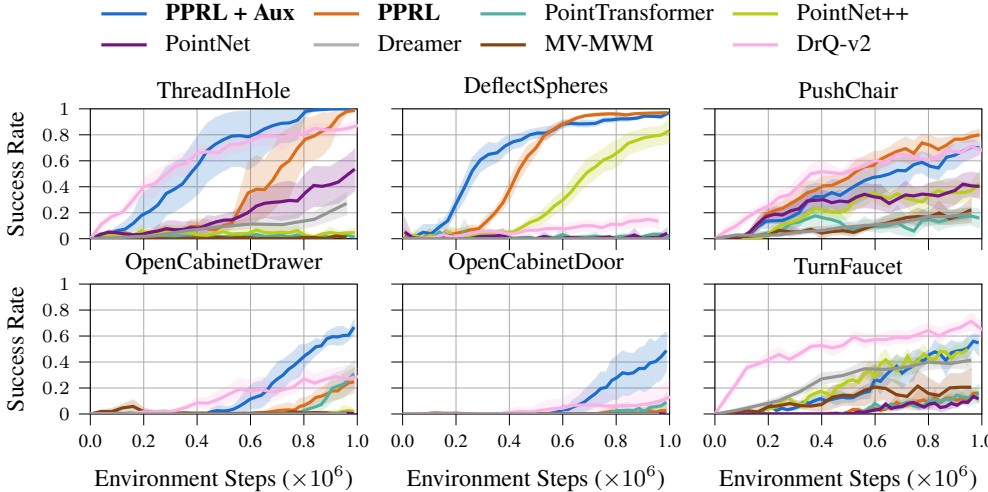

Figure 2: Average success rates and $95\%$ Bootstrap Confidence Intervals [40] for all methods on 2 `sofaenv` and 4 `ManiSkill2` environments. Our method, PPRL + Aux, achieves top performance on 5 of 6 environments. On DeflectSpheres, ThreadInHole, and PushChair, PPRL without representation learning achieves roughly the same success as PPRL + Aux, demonstrating the effectiveness of our neural network architecture even without auxiliary learning objectives. On OpenCabinet-Drawer and OpenCabinetDoor, two challenging tasks with non-trivial variations in scene geometry, adding an auxiliary reconstruction loss significantly improves learning, and is required for solving the task. On TurnFaucet, DrQ-v2 outperforms our method, potentially due to the availability of a static camera, making the learning task easier for image-based methods.

combines random and causal masking as detailed in Appendix B. Finally, a one-layer prediction head reconstructs point patches from the decoded tokens. The loss is the symmetric Chamfer distance [15] between the reconstructed and ground truth patches, which measures the average distance of each point in the ground truth patch to the closest point in the reconstructed patch and vice-versa.

**Variable Point Cloud Size.** While standard representation learning commonly assumes fixed-sized point clouds, in RL, their sizes can vary by orders of magnitude, especially in environments with a moving camera. For point clouds smaller than a fixed minimum size, PointGPT duplicates random points, which may distort objects' point density and add redundant information. To alleviate this problem, we add padding tokens to the input sequence and adjust the masking procedure to maintain a constant ratio of masked and visible tokens among non-padding tokens while ensuring a consistent length for all sequences in a batch.

**Reconstruction of Color Features.** Although most point cloud representation learning methods ignore point-level features such as color, color may be useful in many tasks. We generalize PPRL and PointGPT to include color by adding color features to the points, which are used as inputs by the tokenizer as well as by the reconstruction loss. We will denote $\boldsymbol{x} = [\boldsymbol{x}^{\text{xyz}}, \boldsymbol{x}^{\text{rgb}}]$, $\boldsymbol{x} \in \mathbb{R}^6$ as a single point consisting of the coordinate component $\boldsymbol{x}^{\text{xyz}} \in \mathbb{R}^3$ and the color component $\boldsymbol{x}^{\text{rgb}} \in \mathbb{R}^3$. The color reconstruction minimizes the MSE between the predicted color of each point in the predicted point cloud $\boldsymbol{P}_i^{pd}$ of patch $i$ and the color of the nearest point in the ground truth point cloud $\boldsymbol{P}_i^{\text{gt}}$, i.e.,

$$\mathcal{L}^{\text{rgb}}(\boldsymbol{P}_i^{pd}, \boldsymbol{P}_i^{\text{gt}}) = \frac{1}{|\boldsymbol{P}_i^{\text{pd}}|} \sum_{\boldsymbol{a} \in \boldsymbol{P}_i^{\text{pd}}} \left( \boldsymbol{a}^{\text{rgb}} - \text{nn}(\boldsymbol{a}^{\text{xyz}}, \boldsymbol{P}_i^{\text{gt}}) \right)^2, \tag{1}$$

where $\text{nn}(\boldsymbol{a}^{\text{xyz}}, \boldsymbol{P}_i^{\text{gt}})$ returns the color of the nearest point to $\boldsymbol{a}^{\text{xyz}}$ in the ground truth point cloud $\boldsymbol{P}_i^{\text{gt}}$. These correspondences are already available through the Chamfer distance computation. The final auxiliary loss (referred to as Aux, i.e. PPRL + Aux) is then the sum of Chamfer loss and the color reconstruction loss.

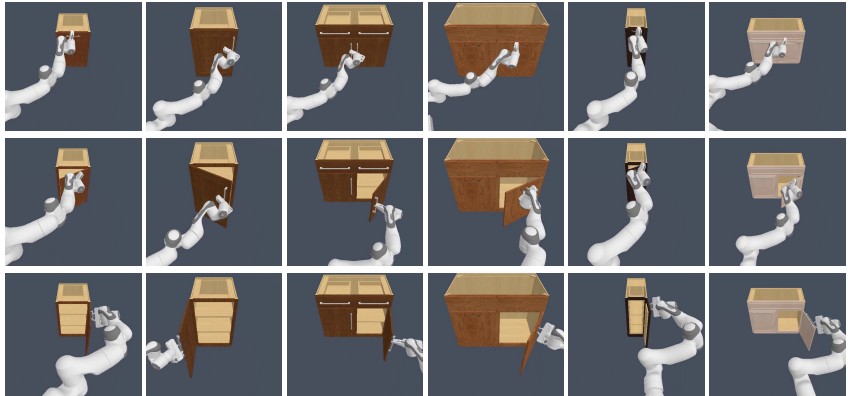

Figure 3: Visualization of successful PPRL + Aux trajectories on the OpenCabinetDoor environment from a static rendering camera that the agent does not have access to. Each column shows a single episode. Agents trained with PPRL + Aux adapt to varying geometries, including handle size and orientation, and whether the door opens to the left or right. Our method is able to coordinate the movements of the gripper and the base and generalize well over these factors.

## 4 Experiments

We compare our method with the auxiliary representation learning loss (PPRL + Aux) and without it (PPRL) on 6 challenging visual robotic manipulation tasks from `sofaenv` [42] and `ManiSkill2` [32]. These environments highlight PPRL's ability to infer non-trivial geometric information from multiple viewpoints and moving cameras. We compare several ablations using different point cloud architectures to showcase the benefits of using point patching with a transformer. Further, we compare against SOTA image-based RL approaches to demonstrate the merits of using point clouds. We report the average success rate and $95\%$ Bootstrapped Confidence Intervals [40]. All results are averaged over $8$ seeds, except for methods that fail[1] where we only run $4$.

**Environments.** `sofaenv` [42] provides environments for robot-assisted surgery, focusing on tasks involving deformable objects. For these tasks, the camera's position and orientation change for each trajectory, mimicking viewpoint changes encountered during surgery with endoscopic cameras. This is exceptionally challenging for most image-based approaches but reflects the desiderata of deploying a learning agent in a real-world system without tedious camera calibration. Furthermore, the deformable nature of the objects involved prevents access to concise state information.

`ManiSkill2` [32] provides challenging robot manipulation tasks, including varying object geometries by sampling object models randomly from PartNet [43] (e.g., cabinets, chairs, faucets, etc.). PushChair, OpenCabinetDrawer, and OpenCabinetDoor are *mobile manipulation* tasks, using 3 cameras mounted above the mobile robot. TurnFaucet exhibits *static manipulation* and uses 2 cameras, mounted to the mobile gripper and the static base. Point clouds are given in the egocentric perspective, which is especially challenging for mobile manipulation, as this reference frame is not static. The agent additionally has access to proprioception and the target location and pose.

We train on ThreadInHole, PushChair, and TurnFaucet without color, while we use color features for DeflectSpheres, OpenCabinetDrawer, and OpenCabinetDoor, where color is either necessary or helpful for detecting the target. Appendix A provides details for all environments.

**Ablations and Baselines.** We compare our encoder against several popular point cloud architectures from the literature. We consider PointNet [12], which has previously been used in the literature for RL on point clouds, as well as its successor PointNet++ [13]. To investigate the benefit of transformer-based architectures, we consider PointTransformer [14].

---

[1] Failing means that none of the $4$ seeds ever reached $> 10\%$ success.

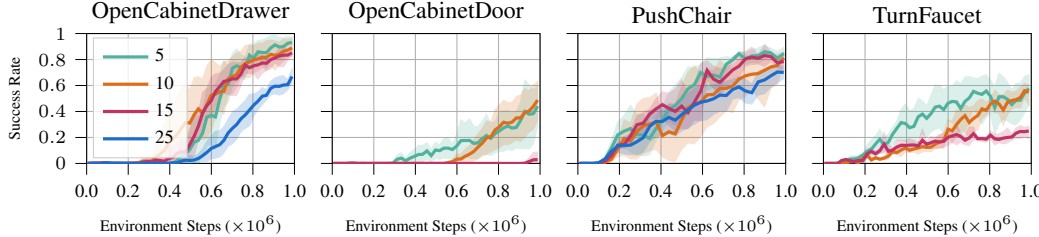

Figure 4: Average success rates and 95% Bootstrap Confidence Intervals for agents trained with PPRL + Aux on selected environments with varying numbers of object models. Although there is a general trend that more object models reduce success rates, the effect is not always strong, showing that the encoder can generalize well. OpenCabinetDrawer performs approximately the same for 5, 10, or 15 object models, and only begins to decrease for 25.

Additionally, we compare PPRL against several SOTA image-based RL approaches, to showcase the benefits of using point clouds. First, we compare against the model-free DrQ-v2 [22] and the model-based Dreamer [7] approaches. To ensure all approaches receive the same information, both get the full RGBD images from all available cameras. Finally, we compare PPRL against Multi-View Masked World Models (MV-MWM) [28], which is tailored to integrate image observations from different camera perspectives and uses a Dreamer-based world model to do sensor fusion in a latent space. Appendix B provides details on the hyperparameters and training of the baselines.

## 4.1 Results

Figure 2 shows the results for all 6 environments tested. PPRL + Aux outperforms all baselines on 5 of 6 environments tested, while DrQ outperforms our method on the TurnFaucet task although the final performance is similar. While PPRL also works on DeflectSpheres, ThreadInHole, and PushChair without representation learning, representation learning leads to substantial benefits in the challenging OpenCabinetDrawer, OpenCabinetDoor, and TurnFaucet tasks. In these tasks, object geometry is crucial for policy execution, showing that representation learning is helpful for the policy to understand the differences in object geometry. For DeflectSpheres and ThreadInHole, representation learning still speeds up learning, while there is a slight decrease in performance on the PushChair task. These tasks either have rather simple object geometries or the difference in the object geometry does not require different strategies for manipulation (PushChair).

PPRL + Aux solves OpenCabinetDrawer and OpenCabinetDoor with success rates of 60% and 50%, respectively, while PPRL without representation learning and the baseline methods are unable to solve them. These environments are challenging mobile manipulation tasks that necessitate adapting the strategy to the target geometry. The handles of each cabinet model have different sizes and poses, and must therefore be grasped differently, and doors must be actuated according to their direction of opening and turning radius. Additionally, all cameras in these environments are mobile, which presents a much more difficult perception problem for image-based methods than for point cloud methods. Figure 3 visualizes the policies for the OpenCabinetDoor environment, showing how the agent responds to variations in the cabinet geometry.

In TurnFaucet, DrQ-v2 has a higher initial success rate than our method and a comparable final success rate. We speculate that this is partly because one of the cameras is static, allowing image-based methods to learn representations without needing to generalize across viewpoints. Dreamer and Multi-View Masked World Models also appear to benefit from the static camera in TurnFaucet. All other environments contain no completely static cameras, either due to variation between episodes or the movement of the robot base, leading to the general tendency for point cloud-based methods to outperform image-based methods. Interestingly, PointNet++ also performs competitively on DeflectSpheres, PushChair, and TurnFaucet, despite its relative simplicity. This may be due to how it processes point cloud data at multiple geometric scales, using neighborhoods generated by FPS to iteratively downsample. Surprisingly, PointTransformer performs poorly across all environ-

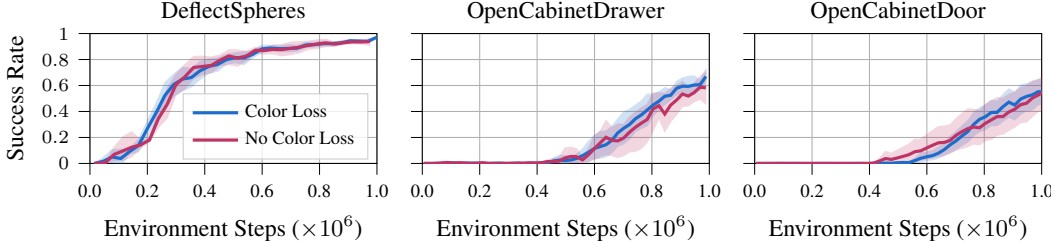

Figure 5: Average success rates and 95% Bootstrap Confidence Intervals for agents trained with and without color reconstruction loss, on all environments with color point cloud observations. When training without color reconstruction loss, color is still observed, but only the signal from RL encourages the agent to condition on color features. In particular the hard Cabinet tasks profit from explicitly reconstructing the color.

ments, despite containing many components common to more successful networks such as PPRL and PointNet++, namely neighborhood computation via FPS and a transformer backbone.

**Object Model Count.** We test the generalization capabilities of our method by training on `ManiSkill2` environments with varying numbers of object models. Figure 4 shows the effect of increasing numbers of object models on the sample efficiency. Although more models generally result in a more difficult learning task, the effect is weak below a certain number of models (15 for OpenCabinetDrawer, 10 for OpenCabinetDoor). This suggests that the encoder's capacity is sufficient for learning more object models, and the limiting factor may rather be the burden of exploring and finding generalizable RL policies with an increasing number of geometries.

**Color Reconstruction.** To investigate the effect of color reconstruction we remove color reconstruction for all 3 environments that use color point clouds. Without the reconstruction loss, only the loss signal from RL encourages the agent to incorporate color into the embedding. The results are shown in Figure 5. Surprisingly, the color reconstruction objective provides no significant benefit for the 3 environments tested, despite the fact that DeflectSpheres encodes some task-relevant information solely through color. Although it is important to provide color to the policy to solve the task, the learning signal from RL alone is enough to learn any color features required. Furthermore, this shows that the Chamfer distance loss is informative enough to learn much of the task-relevant information in the point cloud without any additional modifications.

## 5   Conclusion

We present PPRL, a novel method for RL on point clouds, and an accompanying auxiliary reconstruction objective based on PointGPT. PPRL exploits the inherent properties of point clouds to combine color and positional information from all available cameras and is robust to changing and moving viewpoints. We evaluate PPRL on a series of challenging visual robot (mobile) manipulation tasks from `sofaenv` and `ManiSkill2` and find thatPPRL outperforms popular baselines, even without the auxiliary reconstruction objective. In combination with the auxiliary representation learning objective, our method improves sample efficiency and final success rates in most environments compared to both point cloud-based and image-based baselines. These advantages are more significant for more challenging environments, such as OpenCabinetDrawer and OpenCabinetDoor. A key insight is that point cloud methods offer robustness to changing viewpoints in environments without a static camera, a domain where image-based methods are known to struggle [10].

**Limitations.** While we demonstrated the effectiveness of *existing* representation learning methods for RL on point clouds, findings from image-representation learning with masked reconstruction for RL [27] suggest that special treatment may further improve the performance. While the convolutional feature masking method of [27] is not straightforwardly applicable to point clouds, similar extensions could be investigated. Furthermore, we have not included other sensor modalities such as proprioception in the representation learning, which has shown benefits for image-based RL [44].

**Acknowledgments**

We would like to thank Paul Maria Scheikl for his help with `sofaenv` and the LapGym environment suite. The present contribution is supported by the Helmholtz Association under the joint research school "HIDSS4Health – Helmholtz Information and Data Science School for Health." The authors acknowledge support by the state of Baden-Württemberg through bwHPC, as well as the HoreKa supercomputer funded by the Ministry of Science, Research and the Arts Baden-Württemberg and by the German Federal Ministry of Education and Research.

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

# A Environments

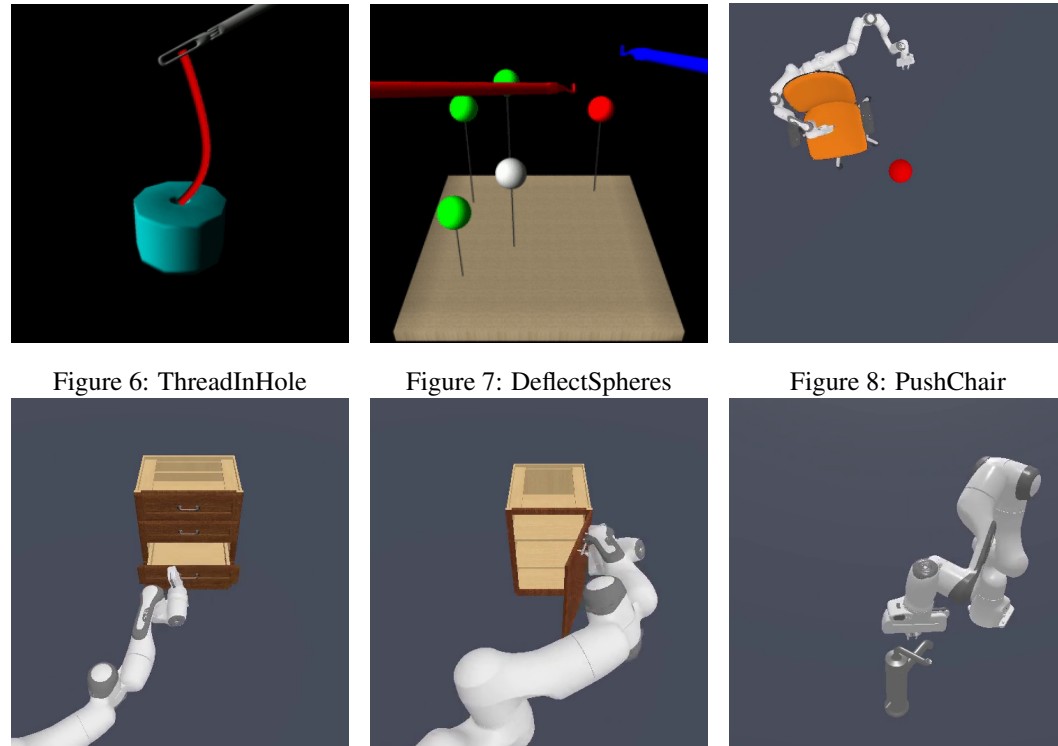

Figure 6: ThreadInHole

Figure 7: DeflectSpheres

Figure 8: PushChair

Figure 9: OpenCabinetDrawer

Figure 10: OpenCabinetDoor

Figure 11: TurnFaucet

Figure 12: Image renderings of the 6 environments used. The `ManiSkill2` environments are rendered from a dedicated, fixed rendering camera that the agent does not have access to.

**Task Descriptions:** All 6 environments are rendered in Figure Figure 12. In ThreadInHole, the goal is to feed the deformable thread into the (rigid) hole. In DeflectSpheres, the goal is to deflect the red or blue sphere with the tool of the corresponding color. Spheres turn green after they are deflected. In PushChair, the goal is to push the chair to the position marked by the red ball (not visible to the agent), without knocking it over. In OpenCabinetDrawer (OpenCabinetDoor), the goal is to grab the drawer (door) handle and open the drawer (door), then stabilize it in the open position. In TurnFaucet, the goal is to turn a faucet to the open position.

`sofaenv`: All `sofaenv` environments have point cloud observations from only a single camera, without a low-dimensional state. Upon each reset, the camera position and look-at are perturbed by a vector uniformly sampled from $[-2cm, 2cm]$ in all directions. Reward weights are taken from the reference implementation.

`ManiSkill2`: All mobile manipulation tasks (OpenCabinetDrawer, OpenCabinetDoor, and PushChair) have 3 cameras mounted above the robot base facing outwards, spaced radially at $120°$. In these tasks, the camera FOV was increased to $1.5$, such that the cabinet/chair is almost always visible from at least one perspective. In OpenCabinetDoor, only the first 10 cabinet models are used for training and evaluation. In addition, the target door is not randomized but fixed for each cabinet model. OpenCabinetDrawer, because it is slightly easier, is not changed, using all 25 models and a randomly chosen target drawer for each episode. TurnFaucet is a static manipulation task with only 2 cameras, one mounted on the (static) base and one inside the gripper. In TurnFaucet, only the first 10 faucet models are used for training and evaluation. In all environments, joint position

delta control is used for the robot arm, and joint velocity control is used for the robot base, if applicable. Observations contain point clouds and a low-dimensional state vector containing robot proprioception and the target location.

**Discounts and Time Limits:** Discounts for all tasks are: 0.99 for ThreadInHole and PushChair, 0.9 for DeflectSpheres, and 0.85 for OpenCabinetDoor, OpenCabinetDrawer, and TurnFaucet. The time limits for all tasks are: 300 for ThreadInHole, 500 for DeflectSpheres, and 200 for all `ManiSkill2` tasks. In OpenCabinetDrawer, OpenCabinetDoor, and TurnFaucet, trajectories are ended only on a timeout, not task success. In ThreadInHole, DeflectSpheres, and PushChair, trajectories are ended either on a timeout or task success. PushChair is the only `ManiSkill2` environment treated like this, because we found that ending on task success is crucial for training.

**Point Clouds:** All environments are rendered with a resolution of 128x128, except PushChair, which has a resolution of 64x64. Egocentric point clouds are created from depth images using the linear transformations given by the camera intrinsic (focal length) and extrinsic (position and orientation) parameters. Point clouds are then post-processed using the following steps: cropping, appending target points, downsampling, and normalization (note that not all environments use all of these steps). First, points beyond the boundaries of the scene are discarded. In `ManiSkill2` environments, the floor is also cropped out, and furthermore in OpenCabinetDrawer and OpenCabinetDoor, any points more than 0.5m in front of the cabinet are also cropped out. In OpenCabinetDrawer, OpenCabinetDoor, and PushChair, 50 additional points are appended to the point cloud, sampled from a uniform cube (of side length 7cm for OpenCabinetDrawer and OpenCabinetDoor and 14cm for PushChair). Although this information is also present in the state vector observation, these points serve to indicate the visual target position within the observation directly. In ThreadInHole, DeflectSpheres, and PushChair, we use voxel grid sampling (with a voxel size of 5mm for `sofaenv` and 5cm for PushChair) to downsample the point clouds, followed by random downsampling (without replacement) to a maximum point cloud size of 1000 points. In OpenCabinetDrawer, OpenCabinetDoor, and TurnFaucet, we randomly downsample each point cloud (without replacement) to a maximum point cloud size of 800 points. For DeflectSpheres, we normalize all observed point clouds by subtracting a fixed center point and dividing by a fixed scale factor. The scale and center point are chosen such that the extrema of a point cloud from a typical initial observation are roughly within $[-1, 1]$. For ThreadInHole, we normalize each observed point cloud independently by subtracting the mean and then dividing by the maximum absolute value over all 3 coordinates, such that the longest axis is mapped to $[-1, 1]$. This method of normalization (compared to that of DeflectSpheres) was crucial for training. Point clouds for `ManiSkill2` environments are not normalized as they already lie roughly within $[-1, 1]$.

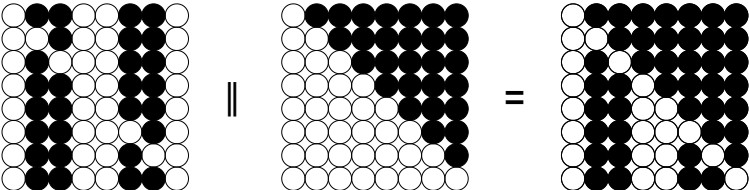

Figure 13: An example of an attention mask for a single token sequence. Black circles are masked (not visible), and white circles are unmasked. The *i*th row shows the tokens that are visible when encoding the *i*th token. The *j*th column shows which tokens may attend to the *j*th token. The first column is the start-of-sequence token, not the token of the first point patch. A token is masked if it is either randomly selected with a probability of $m \in [0, 1]$ (random masking) or is in the present or future (causal masking). A token is always allowed to attend the token immediately preceding it. Additionally, a certain fraction of the tokens at the start of each sequence are never masked out.

## B  Hyperparameters and Architectures

**PPRL**  We use hyperparameters for the tokenizer, masking, encoder, decoder, and prediction head as defined in PointGPT. We reduce the number of transformer layers to 3 to speed up inference. For environments with color, the point feature dimension is set to 3 in the tokenizer and the prediction head; otherwise it is 0. We disable dropout.

**Actor and Critic Networks**  Actor and critic networks are fully-connected networks with 3 layers and 1024 neurons per layer, using ELU as the non-linearity.

**Masking**  Figure 13 provides an overview of the attention mask used for the auxiliary reconstruction loss.

**SAC**  We use SAC with a replay buffer of 500000. We use a replay ratio of 64, except for `sofaenv` environments with a replay ratio of 32, and PushChair where we use 8. We update target networks after every learning update with a $\tau$ of 0.005, except PushChair, which uses a $\tau$ of 0.01. By default, the batch size is 1024 when training without auxiliary loss and 256 when training with auxiliary loss, except for PushChair, where we use 128 in both cases. Due to compute constraints, we reduce the batch size for PointNet++ to 256, and for PointTransformer to 512. Also due to compute constraints, we reduce the replay ratio for PointNet++ on OpenCabinetDrawer and OpenCabinetDoor to 32, as these environments have the largest point clouds on average. The learning rate is 1e-4, except for DeflectSpheres and PushChair with 5e-5, and we use the Adam optimizer. The starting entropy coefficient is 0.1 except for DeflectSpheres with 0.2 and PushChair for 0.05. The learning rate for the entropy coefficient is 1e-4, except for PushChair with 2e-5.

For all baselines, we use hyperparameters as defined by the baselines, except for the environment-dependent discount, initial entropy coefficient, and entropy learning rate, where we use the same values as for our method.

**DrQ-v2**  Each camera uses a dedicated CNN encoder, and the features are concatenated together, resulting in an image encoding of size 117600. Both the image embedding and the state are projected down to a dimensionality of 384 (to match the dimensionality used for PPRL) using a single fully-connected layer and then concatenated.

**Dreamer and Multi-View Masked World Models**  For these world model-based methods, we follow the approach of [45, 4] to handle the additional state information. They first encode both image and state using separate encoders and concatenate their outputs before providing them to the RSSM. For training, they then consider two separate output heads to reconstruct both the image and state. Dreamer uses mostly the hyperparameters proposed by [46]. However, we assume a Gaussian latent variable and use the same discount and entropy control as PPRL. For Multi-View

Masked World Models, we run training in multi-view mode, which masks random viewpoints and reconstructs them. We disable the behavioural cloning loss, as our method does not have access to demonstrations.

## B.1 Model Parameter Counts

Table 1 provides a comparison of the model sizes by listing parameter counts for the proposed method and all ablations. Because the model size changes slightly based on the observation size, and therefore the environment, we use OpenCabinetDrawer for this comparison.

|  | Encoder | Actor | Critic | Total |
|---|---|---|---|---|
| **PPRL+Aux** | 9.47M | 2.91M | 5.80M | 18.18M |
| **PPRL** | 4.95M | 2.91M | 5.80M | 13.66M |
| PointTransformer | 2.20M | 2.91M | 5.80M | 10.91M |
| PointNet++ | 0.38M | 2.91M | 5.80M | 9.09M |
| PointNet | 1.55M | 2.91M | 5.80M | 10.26M |
| DrQ-v2 | 0.08M | 5.08M | 6.25M | 11.39M |
| Dreamer | 2.69M | 0.26M | 0.25M | 3.13M |
| MW-MWM | 4.72M | 0.84M | 0.83M | 6.39M |

Table 1: Comparison of parameter counts for each method.

## C   Wallclock Runtimes

|  | Thread InHole | Deflect Spheres | Push Chair | Open Cabinet Drawer | Open Cabinet Door | Turn Faucet | Average |
|---|---|---|---|---|---|---|---|
| **PPRL+Aux** | 14.1 | 19.3 | 8.7 | 40.9 | 39.3 | 39.6 | 26.0 |
| **PPRL** | 7.4 | 10.7 | 7.4 | 23.2 | 22.5 | 22.3 | 15.0 |
| PointTransformer | 6.6 | 11.9 | 7.8 | 33.0 | 28.5 | 25.7 | 19.3 |
| PointNet++ | 7.7 | 5.9 | 7.3 | 25.1 | 24.1 | 47.7 | 16.4 |
| PointNet | 4.1 | 4.5 | 6.1 | 11.1 | 10.8 | 9.8 | 7.1 |
| DrQ-v2 | 29.4 | 15.0 | 36.5 | 46.2 | 33.5 | 21.5 | 30.3 |
| Dreamer | 25.5 | 15.7 | 28.6 | 27.5 | 26.6 | 17.7 | 24.0 |
| MW-MWM | 32.3 | 13.3 | 23.3 | 22.0 | 21.4 | 14.8 | 21.2 |

Table 2: Comparison of wallclock runtime for each method (in hours).

Table 2 provides a comparison of the average wallclock runtimes for all methods considered. All runs used a single Nvidia A100 GPU, 19 Intel Xeon Platinum 8368 CPUs, and 128Gb of RAM (maximum memory, not average). In some cases, different encoders were run with different hyperparameters such as replay ratio, which has a subsantial effect on the runtime, making direct comparisons difficult. However, we note that image-based methods are among the slowest of the methods we considered, while PointNet is the fastest method by a wide margin. While there are many factors that affect wallclock time, point cloud methods have the advantage that background pixels are already filtered out, which may reduce the compute required for a given observation.

# D   Further Ablations

## D.1   Normalization Type

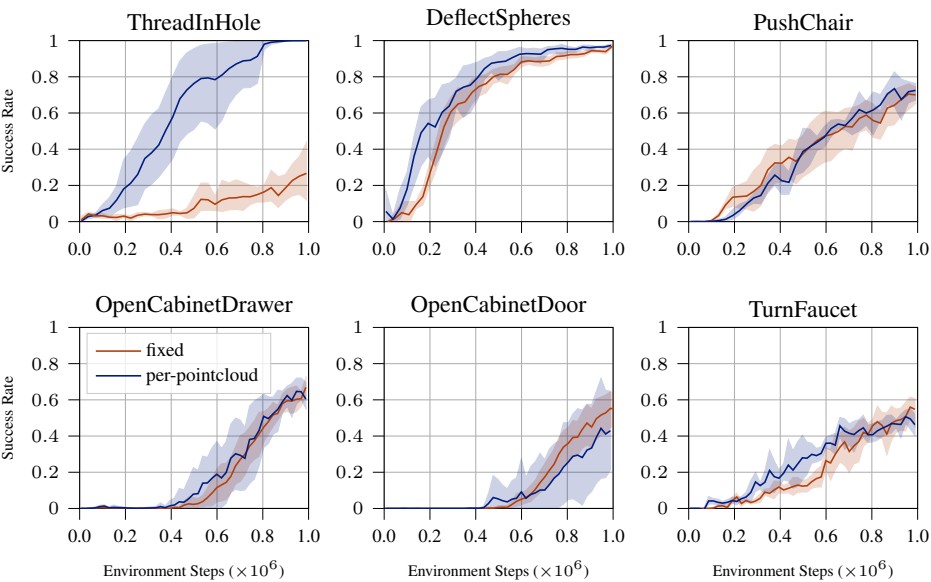

Figure 14:   Average success rates and $95\%$ Bootstrap Confidence Intervals for agents trained with fixed and per-pointcloud normalization on select environments.

To show the effect of different methods for point cloud normalization, we train agents using the opposite normalization type for each environment. Per-pointcloud normalization scales each point cloud independently by subtracting the mean and dividing by the maximum value over all 3 coordinates, such that the longest axis is mapped to $[-1, 1]$. Per-pointcloud normalization is used for ThreadInHole. Static normalization subtracts a fixed center point from each point cloud and divides by a fixed scale factor, where the center and scale are chosen so that initial observations fall roughly within $[-1, 1]$. Static normalization is used for all other environments. A special case of static normalization selects a center at the original and a scale of 1, resulting in no change, which is used for all `ManiSkill2` environments.

Figure 14 shows the effect of normalization type. ThreadInHole depends greatly on per-pointcloud normalization, while other environments appear to be robust to this design choice.

## D.2   Point Patch Size

After a point cloud is tokenized into point patches, the structure of the patches is captured by the positional encoding, while the structure within a point patch can only be captured by the PointNet-based feature extractor. As such, the point patch size determines how fine-grained the agent's perception of the scene is. To investigate the effect of the point patch size (also known as $k$, because it is used in the KNN step), we train PPRL+Aux agents on select environments using $k = [8, 16, 32]$, where $k = 32$ is the default value. The runtime of $k = 64$ is too slow to be practical except for PushChair.

The results of this ablation are shown in Figure 15. We note that learning performance is relatively robust to the value of k in all 3 tasks investigated. We suspect this is because the tokenizer is able to extract all relevant point patch features up to a size of 64. Future work may investigate the optimal trade-off between adding transformer layers and increasing the point patch size.

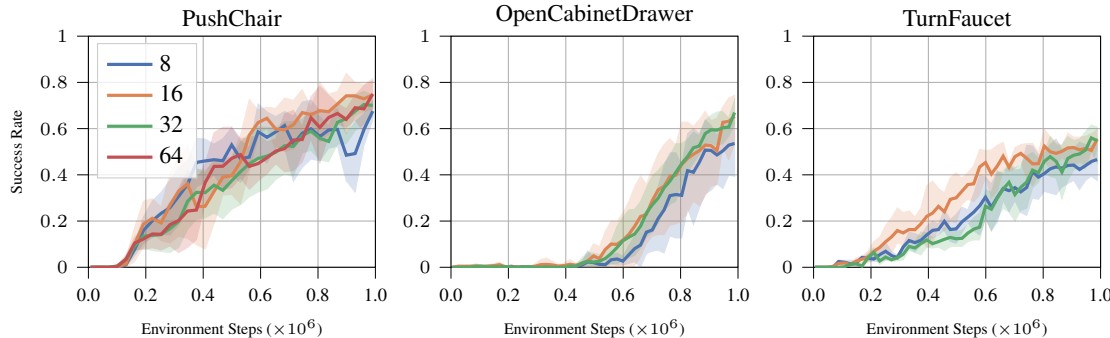

Figure 15: Average success rates and $95\%$ Bootstrap Confidence Intervals for agents trained with varying point patch sizes (k value) on select environments.

