# OpenReview forum: "PointPatchRL - Masked Reconstruction Improves Reinforcement Learning on Point Clouds"
_robot-learning.org/CoRL/2024/Conference — CoRL 2024_

### Official Review · Reviewer_Gb3L · 2024-06-29
**Interesting paper tackling the under-explored problem of 3D representation learning in RL**

**Originality:** 3
**Technical Quality:** 4
**Clarity Of Presentation:** 4
**Potential Impact:** 3
**Recommendation:** 4
**Confidence:** 4

**Review:**

**Strengths**:
* Well-motivated: tackles an interesting representation learning problem for utilizing 3D inputs in RL.
* Conceptually simple (but the technical implementation can get complex).
* Sufficient and convincing experimental benchmark.
* Open-source code (!)

**Weaknesses**:
* Visualizations: it has become common to provide videos of successful and failed rollouts of the learned policies as it is hard to evaluate it visually (e.g., how natural are the learned motions?) from a short sequence of images. However, I couldn’t find any in the supplementary files or in an anonymous webpage (e.g., Google sites).
* Ablations of design choices: as the authors mention in the appendix, the choice of normalization was crucial to get the reported performance. I think a more thorough discussion/comparison of design choices (even if it ends up in the appendix) would complement this paper. For example, choice of PC normalization, pooling mechanism (line 139), masking strategy, number (K) of nearest neighbors and etc…
* Missing training details (See “Questions”).

**Limitations**: Most of the limitations are clearly addressed. A discussion of failure cases would be appreciated as it might help gain insights on further improvements to the proposed approach.

**Quality Of The Limitations Section:**

3

**Questions For Rebuttal:**

* The representation is trained end-to-end on specific tasks. What are the benefits of this approach? Many methods today pre-train a representation model which is then utilized (frozen/fine-tuned) in downstream tasks. Do you think such a 2-stage approach can work in your case (e.g., pre-training your model with the reconstruction loss)?
* The environments in the experiments are simulated and synthetic, how do you think your method would fare with more real-world scenarios where there are also background elements? What would be required to be added to the learning pipeline in order to adapt to such settings?
* If a static camera for image observations seems to provide a useful signal for learning, is it possible to combine image inputs with PPRL to benefit from both worlds?
* Additional training details: what kind of hardware was used to train the models? How long does training take for these computational resources? What are the model sizes?

**Robotics Focus:**

3

**Summary Of Paper:**

Inspired by advances in image representation learning, the paper proposes PointPatchRL (PRRL), a patch-based representation for 3D point-clouds that is learned via Transformer architecture and is further improved by masking parts of the input. The representations are learned end-to-end in downstream reinforcement learning tasks and demonstrate favorable performance in several RL environments compared to other representation approaches.

**Summary Of Recommendation:**

Overall, I am very fond of the ideas presented in this paper, and I largely agree with the claim that working with point clouds in RL is under-explored. I find several key contributions that are novel with moderate impact and significance. I have raised several concerns in the “Weaknesses” and “Questions” parts of my review that I would like the authors to address, but I think the paper is relevant and would be interesting to the CoRL community. As such, my current recommendation is “Weak Accept”, but I’m willing to increase my score given convincing answers to my concerns and the concerns that might be raised by the other reviewers. **Post-Rebuttal**:  I strongly believe that these additions help make the paper clearer and more complete. I am fairly convinced by the authors response to my review and I acknowledge that I have read the other reviews. The other reviewers were concerned by the DrQ baseline and the applicability of the method to real-world settings, which I largely agree with; however, I don't think that it shadows the contribution of this paper as it is focused on robotics, just in simulation. Overall, if all the modifcations and experiments end-up in the revised paper, I believe this paper should be accepted. As such, I'm happy to recommend acceptance..

---

### Official Review · Reviewer_eyuT · 2024-07-19
**Good motivation, need more concrete experiments and fair+stronger baselines**

**Originality:** 2
**Technical Quality:** 2
**Clarity Of Presentation:** 3
**Potential Impact:** 3
**Recommendation:** 2
**Confidence:** 5

**Review:**

**Strengths:**

1. The work investigates an important problem that is very much relevant to the robot learning community on what should be the choice of encoding for point clouds to facilitate sample-efficient RL policies.

2. I appreciate the authors sharing their code for this work. I haven't run it but, I have glanced over the main parts of the code.

**Weaknesses:**

1. Claim that only PointNet has been used: This claim has been made by the authors several times (L35-37, L89-91, L105-107) -- however works like [1] use PointNet+Transformer and [2] use PointConv. Hence, the claim that authors are the first to investigate this (to the best of their knowledge) is incorrect. I'd ask the authors to kindly remove the claim.

2. DrQ-v2 baseline architecture: DrQ-v2, a CNN baseline used for comparison has only a **single** CNN encoder for all the multi-camera images. This is a poor choice of encoder and an unfair baseline. Instead, each camera should have a separate CNN encoder as the filters learned would be specific to each camera.
Additionally, details on how this 117600 dim vector is projected down to 50 are missing. 50-dimensional vector (concatenated with another 50-dim for state encoding) to the policy seems to be a **very small** latent. The typical latent size in literature is of the order of 256/512/1024. I believe the authors need to tune the baseline carefully to have a meaningful comparison and claim that PPRL does better. Given that DrQ-v2 had better sample efficiency with this poor architecture on the TurnFaucet task, I'd expect its performance to significantly increase after accommodating the above-mentioned changes.

3. Instead of only having successful trajectories in Figure 3, I would also like to see the patch-reconstruction quality being evaluated both quantitatively as well as qualitatively in the main paper. Please provide some reconstruction results for PPRL + Aux.

4. Wall clock time analysis: I would expect that point cloud encoders in general to be slower than CNNs for RGB-D images for RL. I would like to see a comparison of wall clock time between **all** the baselines (different point cloud encoders as well as CNNs). This analysis would give a better idea of how much gap well-performing point cloud encoders (such as PointNet++ and PPRL) have compared to PointNet and CNNs.

5. Color Reconstruction experiment: On L277-278, the authors claim that "color reconstruction tends to work better 277 on OpenCabinetDoor" -- I'm not fully on board with this claim. In OpenCabinetDoor, there is a significant overlap of the 95% Bootstrap Confidence Intervals for a policy that uses Color Loss in the Aux. loss and one that does not. So, it is hard to make the claim that color information is significantly helpful.

6. [Very minor comment -- not included to make the decision] I believe the addition of Chamfer loss + color loss is the Auxiliary loss, however, Sec 3.2 that talks about it does not make a reference to the term "Aux. loss" or "Auxiliary loss" and introduction of the word "Auxiliary" directly in Sec 4 (Experiments) is confusing to the reader. Please include a reference to "Auxiliary/Aux." in Sec 3.2.

----

**References**

[1] UniDexGrasp++: Improving Dexterous Grasping Policy Learning via Geometry-aware Curriculum and Iterative Generalist-Specialist Learning, Weikang Wan et al., CVPR 2023

[2] Point Cloud Models Improve Visual Robustness in Robotic Learners, Skand Peri et al., ICRA 2024

**Quality Of The Limitations Section:**

3

**Questions For Rebuttal:**

Experiments to conduct: I would like to see authors address my concerns about the baseline (Weakness #2) and Wall clock time analysis (Weakness #4).

**Additional Question**

7. Have authors experimented with PointTransformer-v2 and v3 and observed any significant differences in the sample-efficient and/or wall clock speed of RL policies.

**Robotics Focus:**

3

**Summary Of Paper:**

This work proposed PointPatchRL (PPRL), an RL method specifically for point clouds that is inspired by prior works that divide the point clouds into overlapping patches and tokenize them. The authors extend this to include masking these patched point clouds and reconstruction -- on the lines of PointMAE (and masked autoencoders in the case of images) specifically for RL. They perform experiments on 5 robotic tasks and show the sample efficiency of the PPRL framework.

**Summary Of Recommendation:**

I am currently voting for a weak reject of the paper on the basis of (a) lack of fair baselines, (b) and incorrect claims regarding contributions. However, I will make my final decision after reading all the other reviews + authors' rebuttal and engaging with authors during the rebuttal.

---

### Official Review · Reviewer_HVuz · 2024-07-23
**It's a well-written paper with a simple but effective approach.**

**Originality:** 3
**Technical Quality:** 4
**Clarity Of Presentation:** 4
**Potential Impact:** 3
**Recommendation:** 3
**Confidence:** 4

**Review:**

Strengths
1. The research direction aligns with the growing interest in using point cloud data in the robotics community, providing a good encoder design choice for processing point cloud data.
2. The paper is well-written and easy to follow. The proposed approach is also straightforward, effective, and technically sound.
3. The authors demonstrated that the proposed method works even when other state-of-the-art image-based approaches struggle in certain tasks, such as ThreadInHole, DeflectSpheres, OpenCabinetDrawer, and OpenCabinetDoor.

Weaknesses
1. The proposed work assumes known camera poses, which is a strong assumption given that it is often hard to obtain camera poses in real-world environments. Depending on applications, visual markers can't be used in real-world environments, so often, one would need to rely on SLAM algorithms to estimate camera poses from real-world videos. This is prone to errors and can negatively affect the registration of point clouds in a common reference frame later. This limitation should be clearly specified, as the paper does not currently address the potential challenges of applying the proposed method in real-world environments.
2. All experiments are only done using simulation data. The paper's claims regarding the use of point cloud representations could have been strengthened significantly with real-world results, especially given the argument in lines 31-33 that there is a potentially smaller sim-to-real gap in point cloud data than in RGB images.
3. In the TurnFaucet result in Figure 2, DrQ-v2 seems to be trained with an additional static camera view. I feel this is not a fair comparison if the proposed method only took an egocentric view as input. The comparison would be fairer if DrQ-v2 were also trained with the same view as the proposed method.
4. The ablation study on a variable number of objects (lines 267-273) is conducted with only two tasks, making it difficult to draw general conclusions. Depending on the task, the effect may be more significant, especially if the task requires understanding interactions between multiple objects. It would be helpful to note this limitation in the ablation study to provide a clearer understanding of its scope.

**Quality Of The Limitations Section:**

2

**Questions For Rebuttal:**

I would like to see answers to 1-3 that I wrote in the weakness section above.

**Robotics Focus:**

3

**Summary Of Paper:**

The paper investigated the use of point cloud representations in training RL learners and demonstrated its effectiveness compared to other image-based representations. The authors adopted a patching-based point cloud representation developed in prior work [15-17] for training RL learners in this paper, where a point cloud is divided into local patches, and a transformer encoder processes each patch as a token. The authors successfully trained this point cloud encoder using the critic's gradients and additionally showed the benefits of training with an additional patch reconstruction loss in challenging simulated robotics tasks.

**Summary Of Recommendation:**

I recommend a weak accept for this paper. Although the paper lacks real-world demonstrations, the findings are still insightful and can lead to more interesting RL papers using point cloud representations in the future. Post-rebuttal: I thank the authors for their efforts in the additional experiments and answers to the reviewers' questions. While I agree with some concerns regarding the DrQ baseline experiment raised by the other reviewer, I am satisfied with the new results the authors provided with the improved DrQ baseline (i.e., the new results didn't change the original claims made in the paper significantly). The strengths I mentioned in my original review remain the same even after checking other reviews and authors' responses, so I will keep my original score.

---

### Author Rebuttal · Authors · 2024-08-09

We thank all the reviewers and the area chair for their time and their valuable comments on our manuscript. We provide a revised version of the manuscript as well as videos of successful and unsuccessful policy rollouts. Changes to the manuscript relative to the submitted version are highlighted in red.

---

### Decision · Program_Chairs · 2024-09-04

**Decision:**

Accept

**Comment:**

PRE REBUTTALS:

High level summary of reviews:

Strengths:

- The research aligns with the growing interest in point cloud data for robotics, offering an effective encoder design choice.
- The paper is clear, easy to follow, and presents a straightforward and technically sound approach.
- The method outperforms state-of-the-art image-based approaches in specific tasks.
- The paper explores an important problem regarding encoding choices for point clouds in RL.
- The method is conceptually simple and demonstrates favorable performance in RL environments.
- Sharing of code.

Weaknesses:

- The assumption of known camera poses is a limitation, as it's often challenging to obtain accurate poses in real-world scenarios.
- The lack of real-world experiments raises questions about the generalizability of the findings.
- The comparison with DrQ-v2 may be unfair.
- The claim that PointNet is the only prior work using point clouds is incorrect.
- The ablation study on the impact of the number of objects is limited to two tasks.
- The lack of visualizations of successful and failed rollouts makes it hard to evaluate the learned policies.
- The paper lacks thorough discussion/comparison of design choices (e.g., normalization, pooling, masking).

POST REBUTTAL:

The authors added additional DrQ-v2 experiments and made a large number of clarifications and modifications to the manuscript during rebuttal. The paper is a stronger submission as a result. There were some concerns raised around the additional DrQ-v2 experiment (pertaining to aggressive dimensionality reduction using only a single linear layer as well as some details around non-shared encoders for multi-view) however, the concerns pertaining to these experiments boiled down to h-parameter choice - which is clearly documented - and so the experiments are a welcome addition. Despite the modifications, reviewer scores were unchanged and still quite high variance post rebuttal. Given the relevance to the community and the additional experiments, this paper meets the threshold for acceptance.